# Effect of Cadmium Chloride and Cadmium Nitrate on Growth and Mineral Nutrient Content in the Root of Fava Bean (*Vicia faba* L.)

**DOI:** 10.3390/plants10051007

**Published:** 2021-05-18

**Authors:** Beáta Piršelová, Emília Ondrušková

**Affiliations:** 1Department of Botany and Genetics, Faculty of Natural Sciences, Constantine the Philosopher University in Nitra, Nábrežie mládeže 91, 94971 Nitra, Slovakia; bpirselova@ukf.sk; 2Department of Plant Pathology and Mycology, Institute of Forest Ecology, Slovak Academy of Sciences, Akademická 2, 94901 Nitra, Slovakia

**Keywords:** cadmium, accompanying anion, fava bean, mineral content, root

## Abstract

The present study aimed to analyze the differences in the tolerance of fava bean (*Vicia faba* cv. Aštar) roots to cadmium in nitrate—Cd(NO_3_)_2_—and chloride—CdCl_2_—solutions. The physiological and biochemical parameters were assessed. The tested doses of Cd (50, 100, 150 and 300 mg/L) did not influence the germination of seeds. However, considerable growth inhibition and dehydration were observed after 96 h incubation. The thickness of roots and rupture of cell membranes increased along with the increasing concentration of the metal in the solution. At a Cd dose of 300 mg/L, irrespective of the solution used, increased nitrogen concentration and no change in sodium content were observed. The content of magnesium increased due to the dose of 100 mg/L (cadmium nitrate) and the content of calcium increased due to the dose of 300 mg/L (in either nitrate or chloride). The correlation analyses pointed to a possible effect of nitrates in the applied solutions on the accumulation of Cd and some minerals in the roots of the given variety of fava bean. This may be important for both research and agricultural practice. The identification of crops with high tolerance to cadmium, as well as knowledge about the mechanisms of ion interactions at the soil solution–plant level, is important in terms of such crops’ use in the process of the remediation of cadmium-contaminated soils coupled with food production.

## 1. Introduction

Among the major environmental pollutants, cadmium (Cd) is one of the most phytotoxic heavy metals. This nonessential element is highly mobile in the soil–plant system and can impair several vital processes, resulting in poor growth and low economic yield of plants [1]. In plants, exposure to Cd causes a delay in germination, stunted growth, chlorosis, alters the chloroplast ultrastructure, inhibits photosynthesis and decreases carbon assimilation by inhibiting enzymes involved in CO_2_ fixation [2,3]. Physiological disorders caused by Cd, including plant biomass reduction, can be an indirect consequence of nutrient deficiencies and the inhibition of photosynthesis [1]. Cd has been shown to interfere with the uptake, transport and use of several elements (Ca, Fe, Mg, P, K, Mn, Cu and Zn) and water by plants [4,5,6]. The mechanism by which Cd inhibits the uptake and utilization of mineral elements is currently not completely clear. It is assumed that Cd may interfere with nutrient uptake by affecting the permeability of the plasma membrane and modifying the activity of the nutrient transporters [7,8], leading to changes in nutrient concentration and composition. For instance, Cd reduced the absorption of nitrate (NO_3_^−^) and its transport from roots to shoots, by inhibiting the nitrate reductase activity in the shoots [9]. There are significant differences in Cd tolerance among species and varieties, but contradictions exist between the results of different experiments. These differences may be attributed to the inherent differential capacity of different species and varieties for Cd accumulation and partitioning in roots and shoots and on the ability to restrict Cd in roots [4].

Recent studies have shown that several important agricultural crops can tolerate certain doses of risk elements in their tissues and can be used for the phytostabilization or phytoremediation of soils contaminated with risk elements. In comparison with other crops (wheat and maize), the fava bean (*Vicia faba* L.) can accumulate and translocate Cd and lead (Pb) in its various tissues. Some genotypes have been suitable for cultivation in Cd lead-contaminated soil without posing a risk to food safety [10]. In addition, the fava bean is an economically important legume crop with high biomass and fast growth, and it is considered an important protein source for the human diet.

Several studies have also shown that Cd accumulation and transport in plants is also dependent on its speciation and bioavailability [11,12,13]. For example, the studies of Hofslagare et al. [14] pointed out that photosynthesis in green alga *Scenedesmus obliquus* was less inhibited by Cd in chloride (Cl^−^)-containing media than in those containing NO_3_^−^.

Both NO_3_^−^ and Cl^−^ are the most abundant inorganic anions in plants and share similar physical properties and transmembrane transport mechanisms, which is the origin of the strong dynamic interactions between the two monovalent anions [15].

Chloride is a major osmotically active solute in the vacuole and is involved in both turgor- and osmoregulation. Chloride also acts as a counter anion, and Cl^−^ fluxes are implicated in the stabilization of membrane potential, the regulation of pH gradients and electrical excitability [16]. On average, a Cl^−^ concentration in the external solution of more than 20 mM can lead to toxicity in sensitive plant species [17].

Nitrate is the form of nitrogen most readily assimilated by plants. It also acts as a signal molecule regulating a wide range of genes and biological processes involved in nitrogen utilization and general plant lateral root development, and it can significantly influence the plant’s tolerance to environmental stresses including Cd toxicity [18,19]. NO_3_^−^ might upregulate the expression of various Cd/Fe transporter genes in plants and further increase Cd accumulation [20]. Nitrogen concentrations exceeding 10 mM NO_3_^−^ generally slow plant growth, while a value of 25 mM is considered the threshold of toxicity for some species [21]. However, several studies have shown inhibition of root growth at doses of nitrogen in the range of 0.1–5 mM [22,23]. Although the effect of accompanying anions in applied Cd solutions on the growth and metabolism of plants is frequent in plant stress physiology, the number of studies focused specifically on the evaluation of the phenomenon has been rather low. Data about the influence of the Cd-accompanying anion on the mineral content in roots during the ontogenesis stage are rather scarce.

Knowledge of ion interaction mechanisms in the soil–plant system can be important in predicting the negative impact of fertilizers or other soil components on crops in areas with elevated cadmium concentrations. The identification of tolerant crops with a high accumulation potential for Cd is also important in terms of their use in the process of the remediation of cadmium-contaminated soils.

The main objectives of this study were (1) to evaluate the toxic effect of Cd on the growth and selected physiological characteristics of fava bean roots and (2) to evaluate possible differences in Cd toxicity when applying different types of salts—Cl^−^ and NO_3_^−^.

## 2. Results

### 2.1. Effect of Cd on Germination of Seeds and Root Growth

The germination percentage of the tested variety was not affected by the applied doses of cadmium nitrate and cadmium chloride (Table 1).

After 96 h of incubation in solutions of Cd, the shortening of roots was observed. These changes were statistically significant, except for the case of the Cd 50 dose (Figure 1A and Figure 2).

The roots were most sensitive to the Cd 300 dose. Cadmium nitrate inhibited the root growth by 54.66% and cadmium chloride inhibited the growth by 56.11%. Visual symptoms of toxicity included not only the shortening of primary roots but also their browning. In both variants of the experiment, its degree progressed as Cd concentration increased (Figure 1A).

The decreases in fresh weight, dry weight and water content were also proportional to the concentrations of applied doses of cadmium. Nitrate decreased the fresh weight by 4.29 to 38.65% and chloride by 9.25 to 47.97% against the control. The decrease in dry weight was the most significant at the Cd 300 dose (Figure 2). The decrease in cell viability was also proportional to the decrease in fresh weight (FW) and dry weight (DW) (Figure 2). The oxidative stress (H_2_O_2_ accumulation) because of all the doses of cadmium (Figure 1B) was confirmed by histochemical staining of roots. The tested variety of fava bean showed, through the determined tolerance index (TI), high tolerance. For each of the used doses of cadmium, TI varied from 95.86 to 77.24 (variant A) and 97.37 to 60 (variant B). In the case of growth parameters of roots (length, FW, DW) and cell viability, no statistically significant differences between variants A and B of the experiment were observed (Figure 2). Shortening of roots due to different concentrations and types of cadmium salts was also accompanied by the thickening of the roots. Due to the doses of cadmium nitrate, the thickness of the roots increased by 12 to 36%, and doses of cadmium chloride caused an increase in root thickness by 0.06 to 24%. The differences within the single doses depending on the applied form of cadmium were observed in the case of the highest dose (Cd 300) (Figure 2).

### 2.2. Effect of Cd on the Mineral Content in Roots

Accumulation of cadmium by the roots was proportional to the dose of the applied metal (Figure 3). In the case of the application of different doses of cadmium nitrate, the amount of cadmium in the roots increased 53.6 to 428.5 times, and in the case of application of cadmium chloride, it increased 32.6 to 435.55 times. Moreover, the doses of cadmium nitrate up to 150 mg/L caused a higher accumulation of cadmium in roots than the doses of cadmium chloride (Figure 3). This difference was statistically significant at doses Cd 50 and Cd 150. Cadmium nitrate caused a moderate increase in iron content in the roots, while cadmium chloride caused slight changes against the control. Accordingly, no changes in the content of sodium were observed (Figure 3). Chloride and nitrate dose of Cd 300 increased the nitrogen content (1.15 times that of the control) (Figure 3). In both variants of the experiment, a gradual increase in the concentrations of magnesium and calcium in the roots as an effect of increasing doses of cadmium was observed (Figure 3). These changes were statistically significant in the variant A Cd 150 dose (1.82 and 1.27 times the control for Ca and Mg, respectively) and in the case of the variants A and B Cd 300 doses (2.78 and 2.25 times the control for Ca and 1.45 and 1.40 times the control for Mg, respectively). The type of cadmium salt applied influenced the Cd content in roots at the Cd 50 and Cd 150 doses, and the Ca content at the Cd 150 dose, where the accumulation of the elements was higher in the roots exposed to the effects of cadmium nitrate (Figure 3).

### 2.3. Correlation Analysis

Pearson’s correlation analysis was carried out to investigate the correlations among Cd uptake and tested physiological parameters (Table 2 and Table 3). Strong negative correlations were found between growth parameters (root lengths, FW, DW and water content) and the dose of applied cadmium. Positive correlations were found between the Cd doses applied, Evans blue uptake and root diameters. Increasing the doses of cadmium chloride and cadmium nitrate caused a gradual increase in the content of Cd (r = 0.97, r = 0.99, *p* < 0.5), Ca (r = 0.95, r = 0.98, *p* < 0.5) and Mg in the roots (r = 0.99, r = 0.98, *p* < 0.5). The content of total nitrogen in the roots depended on the form of the applied cadmium; a gradual increase in the content of nitrogen as a result of the applied dose was observed in the case of nitrate (r = 0.977, *p* < 0.5). A positive correlation was also found between the contents of nitrogen and sodium in the case of the B variant of the experiment (r = 0.965, *p* < 0.5). No correlation between the content of iron and other tested parameters was found (Table 2 and Table 3).

## 3. Discussion

The effect of different concentrations of cadmium chloride and cadmium nitrate on the roots of fava bean was tested. Concentrations of accompanying anions of tested cadmium solutions were 0.890–5.34 M NO_3_^−^ and 0.889–5.33 M Cl^−^. At these concentrations of NO_3_^−^ and Cl^−^, the vast majority of plants show optimal growth, contrary to the cases of the applied dose of cadmium. Only a few studies focus on the toxic effect of cadmium in the context of the influence of the accompanying anions in applied solutions. In addition, almost all of these studies evaluate the issue in the Cd–soil system. Šimek and Tůma [13] evaluated the growth of pea in soil contaminated with Cd(NO_3_)_2_, CdCl_2_ and CdSO_4_. The most noticeable symptoms of toxicity showed a variation with CdCl_2_. Similar results were also observed in the study of Tan et al. [24] with turnip rape (*Brassica campestris*). The formation of CdCl*_n_*^2−*n*^ complexes in the soil solution increases Cd uptake, either by direct uptake of the CdCl*_n_*^2−*n*^ complexes by plants or by increasing the diffusion of Cd^2+^ around the root uptake sites [25,26]. On the other hand, Cd uptake and plant fitness were comparable for CdCl_2_ and CdSO_4_ treatments and depended on the applied Cd concentration [27].

Seed germination and seedling growth are key processes for plant establishment in polluted environments. Treatments (NO_3_^−^ and Cl^−^) with the used Cd concentrations (0–300 mg/L) did not affect the germination of seeds of the tested bean cultivar, which indicates that seed coats can be less permeable to heavy metals following imbibition. Rasafi et al. [28] showed that seed germination was significantly influenced only when concentrations of Cd in the growth medium were relatively high (>500 mg/L), which indicates some resistance of seed germination to the toxicity of these metals. On the other hand, inhibition of fava bean seed germination was observed also at lower doses of cadmium [29], but only in the cases of some varieties, which suggests that the germinating seeds of fava bean showed a differential behavior to cadmium stress concerning cultivars. Kapustka et al. [30] claim that seed germination does not seem to be a sensitive indicator of the phytotoxic effects of Cd, Pb and As in most experimental treatments. No changes were observed in the germination capacity of seeds concerning the applied form of cadmium (Table 1).

In contrast to the effect on germination, the applied doses of cadmium had a negative effect on the root elongation at the dose of 50 mg/L, irrespective of the applied salt type (Figure 1A). Observed blackening of roots (Figure 1A) can indicate metal-induced oxidation of different phenols in roots [31]. The negative effect of Cd on root growth also becomes evident by the accumulation of H_2_O_2_ in root tissues in both variants of the experiment (Figure 1B). Increased H_2_O_2_ accumulation was also observed in bean roots (*Vicia faba*) exposed to doses of 5 and 10 mg/L Cd [32]. Although the negative effect of cadmium on root growth was observed in this study, detected TI calculated from the dry mass of roots (95.86–77.24 in the case of applied cadmium nitrate and 97.37–60.00 in the case of cadmium chloride) suggests high tolerance of the given variety to the doses of Cd. The given variety also showed high tolerance at cultivation in soil substrate enriched with ions of Cd in the dose of 50–100 mg/kg soil [33].

Doses of cadmium (in both applied soil types) also caused thickening of roots and disruption of the roots’ cell membranes directly proportional to the applied dose of metal (Figure 2). Reduced root elongation and disruption of cell membranes exposed to Cd have been observed by several authors [34,35,36]. Other studies have observed no effect of Cd exposure on diameters of bean (*Phaseolus vulgaris* L.) roots or the diameters, lengths or specific surface areas of maize roots [37]. An increase in root diameter and reduced root elongation is a morphological adaptation to drought stress as a response to the lower permeability of the dried soil [38]. A similar adaptation occurs also in the event of heavy metal stress [5,39].

Content of cadmium accumulated in roots was higher in cases of the application of cadmium nitrate in the dose range of Cd 50–150 mg/L (Figure 1). Several studies confirmed the effect of NO_3_^−^ on the intake and accumulation of cadmium in roots and the effect of cadmium on the metabolism of nitrogen. The results of these studies are often contradictory. The synergistic effect of nitrogen and cadmium was observed, for example, in rice [20], chamomile [40] and wheat [41]. No statistical differences were seen between the root and shoot N contents of Cd-treated plants relative to the controls, regardless of the Cd doses used [42]. On the other hand, Cd inhibited NRT1.1-mediated NO_3_^−^ uptake in *Arabidopsis* and *Brassica* [43,44,45]. Studies with *Synechocystis aquatilis* algae have shown reduced Cd uptake in nitrates and chlorides, with less Cd accumulated by algae in chlorides [11]. Unlike in soils, the formation of CdCl*_n_*^2−*n*^ complexes decreases Cd^2+^ availability in the nutrient solution [26]. It is possible that the lower accumulation of Cd by the roots in the environment of Cl^−^ ions was in the dose range of 50–150 mg/L due to the lower bioavailability of Cd. Several studies have shown reduced accumulation of Cd by tissues due to increased salt (NaCl) concentration in the nutrient solution [46]. Recently, Souguir et al. [47] examined the effect of combined cadmium (0.01 mM cadmium nitrate), salinity (50 and 150 mM NaCl) and stress on growth in *Vicia faba*. Growth parameters (root length and fresh and dry matter), mitotic activity and micronucleus formation were not influenced by Cd or a low concentration of NaCl when applied separately or together, while 150 mM of NaCl, alone or combined with Cd, negatively affected all the studied parameters. Therefore, cadmium toxicity is a function of its speciation and the concentration of particular chemical forms in solutions.

Statistically significant positive correlations were observed between the content of cadmium and content of Mg and Ca in roots of both variants of experiments (Table 2 and Table 3). Increased contents of Ca and Mg in roots due to Cd toxicity were also observed by other authors [48]. Calcium (Ca) is an essential plant macronutrient that is involved in various plant physiological processes, such as plant growth and development, cell division, cytoplasmic streaming and photosynthesis. Calcium, as the second messenger, is involved in signaling nutrient availability and changes thereof [49]. As stated earlier, NO_3_^−^ treatment rapidly increased the cytoplasmic Ca^2+^ level in the roots [50]. Authors of several studies have pointed to the protective role of Ca and Mg in plant development under heavy metal conditions [51,52]. Due to the chemical similarity between Ca and Cd (similar charge and ionic radius), Ca may also mediate Cd-induced physiological or metabolic changes in plants. There exists evidence that cadmium permeates the cytosol through calcium channels in the plasmalemma, resulting in changes in the cell–water relationship [53].

Contents of iron and sodium were not dependent on the applications of the given doses of cadmium, although competition between Fe and Cd in their transport across membranes was confirmed [8]. In the study of Gomes et al. [42], Fe content did not statistically differ between Cd-treated and control plants (except at 25 µmol Cd). Muradoglu et al. [48] found that supplementation of Cd increased K, Mg, Fe, Ca, Cu and Zn concentration in both roots and leaves of strawberry. Several studies, however, pointed to a decrease in the contents of mineral elements because of cadmium [54,55,56,57]. These inconsistent results are probably due to differences in the plant species, organs and/or assay methods used in various studies. During the incubation of germinating seeds in distilled water (without the addition of nutrients), changes in the contents of macro- and micronutrients in roots—because of changes in uptake from the environment—were not considered. The above-mentioned differences in the contents of microelements can also be a result of the fact that the microelements participate in different (unequal) shares on the germination capacity of seeds [54]. The reaction of germinating seeds to cadmium ions is a complex, phytohormone-regulated interplay of various membrane transporters that is likely to be determined by the genotype as well as by the concentration of the applied metal and its speciation [11,58].

It can be concluded that the observed changes in the growth parameters and contents of mineral elements in roots are possibly a direct result of the interaction of ions in the applied solutions, tissue dehydration due to metal exposure or the effects of different shares of the microelements on the germination capacity of seeds. The results also showed a high tolerance of the tested bean variety’s roots to cadmium ions, regardless of the applied form of metal. However, the accumulation of Cd in roots was higher in the dose range of 50–150 Cd in the presence of NO_3_^−^ ions.

## 4. Materials and Methods

### 4.1. Plant Material and Growth Conditions

Seeds of fava bean (*Vicia faba* cv. Aštar) were surface-sterilized with 5% (*v*/*v*) sodium hypochlorite for 5 min, then rinsed five times in distilled water. Afterwards, the seeds were soaked for 12 h in distilled water at room temperature and transferred to Petri dishes (Ø 15 cm) lined with filter paper (Whatman No. 1) moistened with 30 mL of distilled water (control) and 30 mL of solutions of Cd(NO_3_)_2_·4H_2_O (variant A) or CdCl_2_·2H_2_O (variant B). The used concentrations of Cd^2+^ were 50, 100, 150 and 300 mg/L (Cd 50, Cd 100, Cd 150 and Cd 300). Seeds of each variant of the experiment for each Cd level were germinated in two Petri dishes (2 × 30 seeds) in the dark at 25 °C. Each experiment variant was repeated three times.

### 4.2. Determination of Seed Germination

Seed germination was determined after 96 h of incubation in water or metal solutions. Seeds with roots longer than 3 mm were considered germinated. In each variant, 85 seeds were analyzed. The seed germination percentage was calculated using the following formula:

Germination % = Number of germinated seeds/Total number of seeds × 100

### 4.3. Measurement of Growth Parameters

After washing, the fresh weight (FW) of roots was measured and the roots were oven-dried at 60 °C for 48 h to constant dry weight (DW). The water content was calculated as the difference between FW and DW. Tolerance index (TI) was calculated as a ratio of the mean dry weight of plants grown in the presence of cadmium and the mean dry weight of control plants expressed as a percentage [59]. Hand-made cross-sections of primary roots taken at 1 cm distance from the root tip were investigated microscopically (bright field, Axioskop 2 plus, Carl Zeiss, Göttingen, Germany) and photographed with a Kappa 1300 digital CCD camera (Kappa GmbH, Gleichen, Germany). Three replicates per treatment were used. Eight to ten plants from each Petri dish were analyzed (altogether 24–30 plants). For root diameter determination, 40 sections from eight different roots were analyzed in each variant of the experiment.

### 4.4. Determination of Cell Viability

The loss of cell viability was evaluated by spectrophotometric assay of Evans blue staining [60]. Roots were stained in a 0.25% (*w/v*) aqueous solution of Evans blue for 15 min at room temperature. The stained roots were washed three times with distilled water for 5 min. Five root tips (approximately 1 cm long, 300 mg) were excised and soaked in N, N-dimethylformamide for 1 h at room temperature. Absorbance was measured spectrophotometrically at 600 nm (spectrophotometer UV-2601, Shimadzu, Japan). The absorbance of samples is proportional to the degree of cell damage.

### 4.5. Visualization of H_2_O_2_ with the Diaminobenzidine (DAB) Method

Hydrogen peroxide (H_2_O_2_) was detected with 3,3′-diaminobenzidine tetrachloride (DAB) reagent (Sigma-Aldrich, Darmstadt, Germany) [61]. Roots were immersed in 0.05% DAB (pH 3.8) for 3 h at room temperature in the dark. Subsequently, roots were washed in distilled water and photographed.

### 4.6. Detection of Mineral Elements in Roots

Dried and ground roots from each repetition of the experiment (pooled from two Petri dishes) were used to detect mineral elements. Plant material (0.2 g) was digested in the mixture of 5 mL water, 5 mL of concentrated HNO_3_ p.a. (Merck, Darmstadt, Germany) and 1.5 mL of H_2_O_2_ p.a. (Slavus, Bratislava, Slovakia) by using the Mars Xpress microwave oven (*CEM* Corporation, Matthews, NC, USA). The parameters of the decomposition process were as follows: temperature 140 °C, ramp time 15 min and hold time 13 min. After digestion, the solution was diluted to 25 mL with deionized water and filtered through an acid-resistant cellulose filter (Whatman No. 42). Blank samples were prepared identically. Cadmium (Cd) was determined by flame atomic absorption spectrometry (AAS-FAAS) using the Thermo Scientific series 3000 (Shanghai, China). Elementary calcium (Ca), magnesium (Mg), iron (Fe) and sodium (Na) were detected using inductively coupled plasma optical emission spectroscopy (ICP-OES 725, Varian 725 ES ICP, Melbourne, Australia). The Flash EA 1112 Elemental Analyzer (Thermo Finnigen, Milan, Italy) was used to determine total nitrogen (N) in samples.

Limits of quantification (LOQ) and limits of detection (LOD) for elements were: 100 and 30 mg/kg for Ca, 40 and 12 mg/kg for Mg, 4 and 1.2 mg/kg for Fe, 20 and 6 mg/kg for Na, 0.5 and 0.15 mg/kg for Cd and LOD 0.1 g/kg for N. The legitimacy of the whole method was verified using the certified reference materials: RM 100 Wepal for N and NIST 1575a Pine needles for Cd, Ca, Mg, Fe and Na.

### 4.7. Statistical Analyses

Obtained data were processed using the statistical package of MS Excel. The significance of differences was analyzed by using a Student’s *t*-test, and *p* < 0.05 was considered as statistically significant. Pearson’s correlation coefficient was calculated between tested parameters.

## 5. Conclusions

The roots of a given variety of fava bean showed relatively high tolerance to cadmium ions. Reductions in root growth, as well as changes in mineral content, were observed at higher doses of Cd (150 and 300 mg/L).

The increased concentrations of Mg and Ca in the roots point to their potential protective role in the conditions of cadmium exposure. Although our experiments did not clearly demonstrate the possible toxic effect of Cl^–^ and NO_3_^–^ ions on the monitored parameters, it was shown that the accumulation of Cd and some minerals might depend on the concentration of the metal in applied solutions as well as on its speciation.

This finding may be important in predicting the risks of fertilizers affecting plants in cadmium-contaminated soils. Another important aspect is the study of cadmium toxicity mechanisms in the context of interaction with other ions in the environment.

## Figures and Tables

**Figure 1 plants-10-01007-f001:**
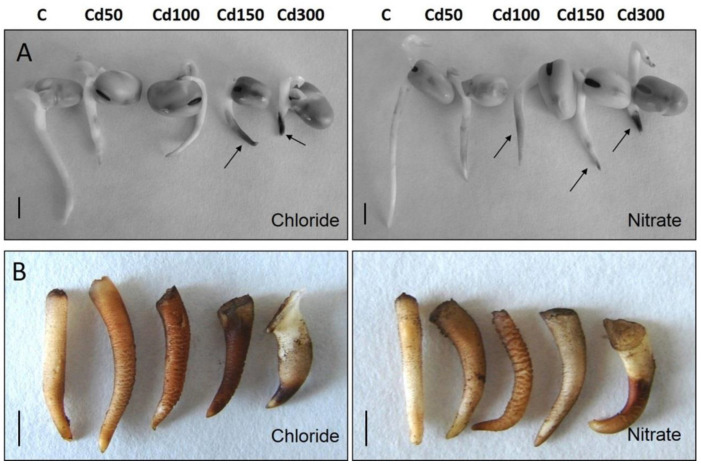
Effects of different doses of cadmium (Cd) (mg/L) on the growth (**A**) and oxidative damage (**B**) of roots of fava bean. Arrows indicate the blackening areas of the plant tissue. Brown color shows H_2_O_2_ accumulation. C—Control, Scale bar = 1 cm.

**Figure 2 plants-10-01007-f002:**
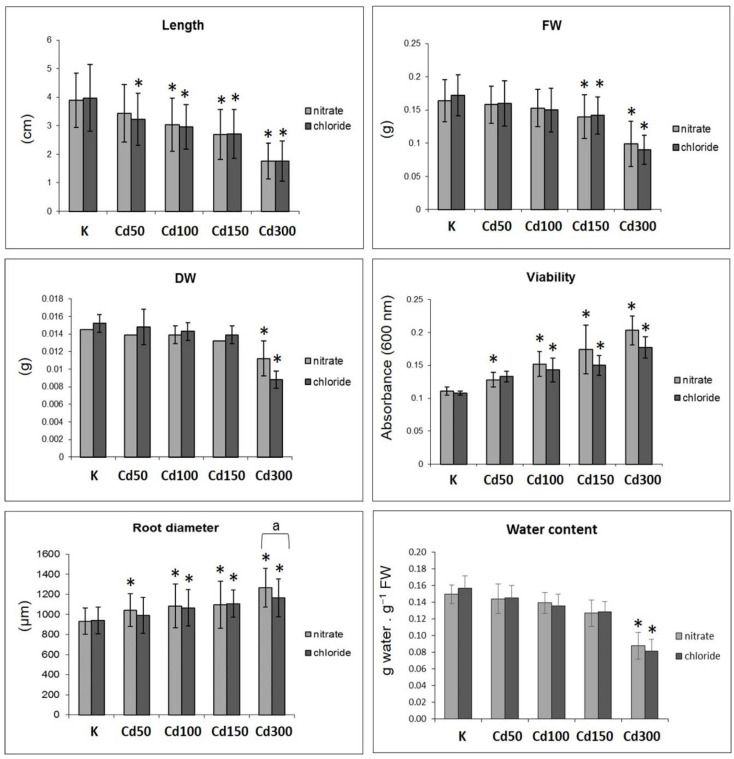
Effects of different doses of cadmium (0–300 mg/L) applied in the form of nitrate or chloride on the growth and physiological parameters of roots of fava bean. The values represent the arithmetic mean ± standard deviation. *—the level of significance of the differences against the control (0 mg/L Cd) at *p* < 0.05, a—statistically significant differences between the variants of the experiment (nitrate and chloride) at *p* < 0.05. FW—fresh weight, DW—dry weight.

**Figure 3 plants-10-01007-f003:**
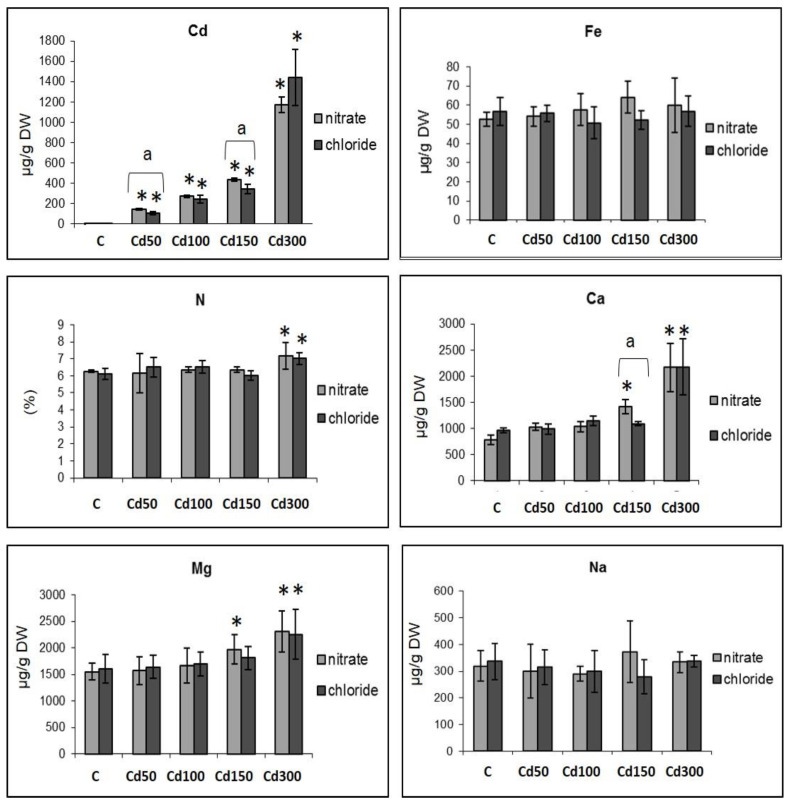
Effects of different doses of cadmium (0–300 mg/L) applied in the form of nitrate or chloride on the content of cadmium (**Cd**) and minerals—iron (**Fe**), nitrogen (**N**), calcium (**Ca**), magnesium (**Mg**) and sodium (**Na**)—in roots of fava bean. The values represent the arithmetic mean ± standard deviation. *—the level of significance of the differences against the control (0 mg/L Cd) at *p* < 0.05; a—statistically significant differences between the variants of the experiment (nitrate and chlo-ride) at *p* < 0.05.

**Table 1 plants-10-01007-t001:** Germination percentage (%) of *Vicia faba* seeds after imbibition with water H_2_O (0) and different concentrations of Cd.

Applied Form of Cd^2+^	Concentration of Cd^2+^ (mg/L)				
	0	50	100	150	300
Nitrate	100	97	98	100	98
Chloride	99	100	99	98	99

**Table 2 plants-10-01007-t002:** Correlation matrix between cadmium doses applied, growth parameters, Evans blue uptake, mineral and water content in roots after applying cadmium nitrate to the growth medium.

	Cd Doses	RL	FW	DW	EB	RD	Cd	Fe	N	Ca	Mg	Na	H_2_O
Cd doses	1												
RL	−1.00	1											
FW	−0.99	0.98	1										
DW	−0.97	0.95	0.99	1									
EU	0.97	−0.98	−0.94	−0.88	1								
RD	0.99	−0.98	−0.99	−0.99	0.92	1							
Cd	0.99	−0.98	−1.00	−0.99	0.93	0.99	1						
Fe	0.52	−0.56	−0.44	−0.31	0.38	0.38	0.42	1					
N	0.98	−0.97	−1.0	−0.98	0.91	0.99	0.99	0.34	1				
Ca	0.98	−0.98	−0.99	−0.98	0.93	0.98	0.98	0.47	0.96	1			
Mg	0.981	−0.98	−0.97	−0.93	0.99	0.94	0.96	0.65	0.92	0.98	1		
Na	0.46	−0.48	−0.42	−0.33	0.59	0.32	0.38	0.90	0.26	0.49	0.62	1	
H_2_O	−0.99	0.98	1.00	0.99	−0.93	−0.99	−0.99	−0.43	−0.98	−0.99	−0.97	−0.42	1

Grey cells indicate a significant correlation at the 0.5 level. RL—root length, FW—fresh weight, DW—dry weight, EB—Evans blue uptake, RD—root diameter, cadmium (Cd), mineral (Fe, N, Ca, Mg, Na) and water (H_2_O) contents in roots.

**Table 3 plants-10-01007-t003:** Correlation matrix between doses applied, growth parameters, Evans blue uptake, mineral and water content in roots after applying cadmium chloride to the growth medium.

	Cd Doses	RL	FW	DW	EB	RD	Cd	Fe	N	Ca	Mg	Na	H_2_O
Cd doses	–												
RL	−1.00	–											
FW	−0.99	0.99	–										
DW	−0.97	0.98	1.00	–									
EU	1.00	−1.00	−0.99	−0.97	–								
RD	0.95	−0.94	−0.89	−0.85	0.95	–							
Cd	0.97	−0.98	−1.00	−1.00	0.98	0.86	–						
Fe	0.45	−0.48	−0.55	−0.61	0.44	0.18	0.59	–					
N	0.58	−0.62	−0.69	−0.75	0.59	0.33	0.74	0.65	–				
Ca	0.95	−0.96	−0.98	−0.99	0.95	0.81	0.99	0.59	0.81	–			
Mg	0.99	−0.99	−1.00	−0.99	0.99	0.89	0.99	0.58	0.66	0.97	–		
Na	0.52	−0.56	−0.64	−0.71	0.53	0.23	0.70	0.81	0.96	0.75	0.63	–	
H_2_O	−0.99	1.00	1.00	0.99	−0.99	−0.90	−1.00	−0.54	−0.68	−0.98	−1.00	−0.63	–

Grey cells indicate a significant correlation at the 0.5 level. RL—root length, FW—fresh weight, DW—dry weight, EB—Evans blue uptake, RD—root diameter, cadmium (Cd), mineral (Fe, N, Ca, Mg, Na) and water (H_2_O) contents in roots.

## Data Availability

The data presented in this study are available on request from the corresponding author.

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
