# Peer review of "Effect of Cadmium Chloride and Cadmium Nitrate on Growth and Mineral Nutrient Content in the Root of Fava Bean (Vicia faba L.)"

_plants, 2021, doi:10.3390/plants10051007_

Round 1
Reviewer 1 Report
The manuscript entitled" Effect of cadmium chloride and cadmium nitrate on growth and mineral nutrient content in the root of fava bean (Vicia faba ) is an interesting article too read—however the material and methods were not clearly written and needs major changes before further perusal—
Abstract – well written
Introduction- sufficiently written and described
Results – sufficiently described
Discussion – properly drawn
Material and methods-
- Line no- 282- suggestion- its not right to write growth media. As these are the treatments which authors have applied, I suggest please change the sentence –
- Please specify what quantity (ml /per petridish) of different concentrations of Cd was supplied to each Petri-dish, along with control distill water.
- There was no mention found in the methodology that how many seeds were kept for germination in per Petri-dish and replications of the treatment. I suggest please include all these in detailed--
- Line no- 288- After 96 h of plant growth—please change the sentence as in 96 hours it can't be plant growth as seeds have germinated and it must be still in seedling stage – Please rectify the sentence—
- Section 4.2 Determination of seed germination and measurement of growth parameters—in this section authors performed experiments for % germination and growth parameters, however its hard to understand why they performed Hand-made cross-sections of primary roots, if it belongs to growth parameter, please include a subheading to make it more clearer—
- Authors performed cell viability test however no figures were provided along with comparisons , it is requested to please include the figures for the same
- Section 4.5- Authors say that they have dried plant material (0.2g) – but when the experiment was conducted only for 96 hours how it become plant – its hard to understand please rewrite and also clarify how many repetitions ( for a single treatment) were of the Petri-dishes were kept so that enough material was obtained for elemental analysis, secondly also clarify if the materials were pooled after drying from diffrent petridishes or they were processed individually—
Conclusions – its too descriptive, I suggest to authors please concise it and include only a paragraph about what conclusion they obtained from the study rather than describing in details about results which they have alwready presented in result section--
Author Response
Dear Reviewer 1.
Thank you for the comments. I am sending you the revised version of the manuscript entitled “Effect of cadmium chloride and cadmium nitrate on growth and mineral nutrient content in the root of fava bean (Vicia faba L.).” We have considered all comments of the reviewers in the manuscript. All of modifications were marked by red color in revised manuscript. However, if there is any other comment for improving the MS I would be much appreciated to know.
Best Regards,
Emília Ondrušková
Response to Reviewer 1:
Comment: Line no- 282- suggestion- its not right to write growth media. As these are the treatments which authors have applied, I suggest please change the sentence –
Response 1: The sentence was corrected as recommended by reviewers. Line 306.
Comment: Please specify what quantity (ml /per petridish) of different concentrations of Cd was supplied to each Petri-dish, along with control distill water.
There was no mention found in the methodology that how many seeds were kept for germination in per Petri-dish and replications of the treatment. I suggest please include all these in detailed—
Response 2: The methodology was corrected according to the opponents' comments. The number of seeds per treatment, the amount of medium applied and the number of replicates of the experiment were added. Lines 303-347.
Comment: Line no- 288- After 96 h of plant growth—please change the sentence as in 96 hours it can't be plant growth as seeds have germinated and it must be still in seedling stage – Please rectify the sentence—
Response 3: The sentence was corrected as recommended by reviewers. Line 306.
Comment: Section 4.2 Determination of seed germination and measurement of growth parameters—in this section authors performed experiments for % germination and growth parameters, however its hard to understand why they performed Hand-made cross-sections of primary roots, if it belongs to growth parameter, please include a subheading to make it more clearer—
Response 4: Section 4.2 was divided to Sections 4.2 (Determination of seed germination) and 4.3 (Measurement of growth parameters), which contributed to the clarity. Lines 305-322.
Comment: Authors performed cell viability test however no figures were provided along with comparisons , it is requested to please include the figures for the same
Response 5: Cell viability is explained in Figure. 2. Evans blue dye accumulates in cells with impaired membrane integrity. The degree of membrane damage was expressed indirectly as the amount of dye extracted from the samples - proportional to the cell damage. There is a negative correlation between viability and absorbance of samples (the higher the absorbance - the lower the viability). We have added this information to section 4.4. Lines 330-331.
Comment: Section 4.5- Authors say that they have dried plant material (0.2g) – but when the experiment was conducted only for 96 hours how it become plant – its hard to understand please rewrite and also clarify how many repetitions ( for a single treatment) were of the Petri-dishes were kept so that enough material was obtained for elemental analysis, secondly also clarify if the materials were pooled after drying from diffrent petridishes or they were processed individually—
Response 6: Dried and ground roots from each replicate of the experiment (pooled from two Petri dishes) were used for detection of mineral elements. We present this fact in section 4.5. The values of FW and DW are shown in the graph (Figure 2). They represent average amounts per root. The required amount of material for each variant (0.2 g DW) was obtained in each repetition from approx. 20-40 roots. Lines 346-347.
Comment: Conclusions – its too descriptive, I suggest to authors please concise it and include only a paragraph about what conclusion they obtained from the study rather than describing in details about results which they have alwready presented in result section--
Response 7: The Conclusions were rewritten according to the instructions of the reviewers.
Reviewer 2 Report
Authors aimed to compare the tolerance of fava bean roots to cadmium in nitrate and chloride solutions. In the Abstract the exact aim is not so written, I think it is too long to read the all measured parameters. Overall, Abstract should be rewritten I order to reflect the aims of this MS.
In the Introduction, abbreviations should be used if a word appeared in the manuscript, eg. cadmium or nitrate which is missing from the text.
Materials and methods contain assays which are not correctly give information about the Cd induced responses. For example, H2O2 was just visualized by a staining method. Authors should make some quantitative analysis eg. by spectrophotometer in order to gain information about oxidative stress responses. Also, other ROS should be analyzed. The diagrams should be reorganized by a new statistical approaches as the present form is not correct. Both nitrate and chloride solutions should be analyzed statistically sperarately eg. big or small letters, respectively.
In the Discussion, some results are compared to data from literature involving other stress types, eg. in the case of L 222 on the P8, it is not correct to compare drought stress these results.
There is a lack about the investigation of transporters of cadmium at the level of gene expression. It would be nice to see how these genes are involved in these responses.
Author Response
Dear Reviewer 2.
Thank you for the comments. I am sending you the revised version of the manuscript entitled “Effect of cadmium chloride and cadmium nitrate on growth and mineral nutrient content in the root of fava bean (Vicia faba L.).” We have considered all comments of the reviewers in the manuscript. All of modifications were marked by red color in revised manuscript. However, if there is any other comment for improving the MS I would be much appreciated to know.
Best Regards,
Emília Ondrušková
Response to Reviewer 2:
Comment:
Authors aimed to compare the tolerance of fava bean roots to cadmium in nitrate and chloride solutions. In the Abstract the exact aim is not so written, I think it is too long to read the all measured parameters. Overall, Abstract should be rewritten I order to reflect the aims of this MS.
Response 1: The Abstract chapter was corrected according to all reviewers comments, concluding point was added.
Comment:
In the Introduction, abbreviations should be used if a word appeared in the manuscript, eg. cadmium or nitrate which is missing from the text.
Response 2: In the introduction chapter, abbreviations were used instead of words. They are marked in red. Lines 42, 60, 61, 72, 74, 78, 80.
Comment:
Materials and methods contain assays which are not correctly give information about the Cd induced responses. For example, H2O2 was just visualized by a staining method. Authors should make some quantitative analysis eg. by spectrophotometer in order to gain information about oxidative stress responses. Also, other ROS should be analyzed. The diagrams should be reorganized by a new statistical approaches as the present form is not correct. Both nitrate and chloride solutions should be analyzed statistically sperarately eg. big or small letters, respectively.
Response 3: Histochemical detection of H2O2 is an additional analysis confirming damage of the roots exposed to Cd (viability) ion. Increased accumulation of H2O2 was also confirmed in the roots of beans (Vicia faba) exposed to doses of Cd 5 and 10 mg / L, which we also pointed out in the discussion. A new citation has been added to the manuscript.
Lin AJ, Zhang XH, Chen MM, Cao Q.: Oxidative stress and DNA damages induced by cadmium accumulation. J. Environ. Sci. (China) 2007,19 (5), 596-602. doi: 10.1016/s1001-0742(07)60099-0. doi: 10.1016/s1001-0742(07)60099-0.
Both nitrate and chloride solutions were statistically analyzed separately, with statistically significant changes relative to control within each variant (chloride and nitrate). Also at each dose, the sets between chloride and nitrate data were compared to express differences in their effect (Fig. 2 and 3): "* - the level of significance of the differences against the control (0 mg / L Cd) at p <0.05, a - statistically significant differences between the data (nitrate and chloride) at p <0.05." Pearson correlation matrix was made on the basis of comparison of the relative values of recorded traits under cadmium chloride (Table 2) and cadmium nitrate conditions (Table 3). This method of expressing relationships between parameters has also been used by other authors (Pailles et al., 2020; Quijada et al., 2020).
Pailles Y, Awlia M, Julkowska M, Passone L, Zemmouri K, Negrão S, Schmöckel SM, Testera M.: Diverse traits contribute to salinity tolerance of wild tomato seedlings from the Galapagos Islands. Plant Physiology 2020, 182, 534–546.
Quijada NM, Schmitz-Esser S, Zwirzitz B, Guse CH, Strachan CR, Wagner M, Wetzels SU, Selberherr E.: Dzieciol, M.: Austrian raw-milk hard-cheese ripening involves successional dynamics of non-inoculated bacteria and fungi, Foods 2020, 9, 1851; doi:10.3390/foods9121851
Comment:
In the Discussion, some results are compared to data from literature involving other stress types, eg. in the case of L 222 on the P8, it is not correct to compare drought stress these results.
Response 4: Exposure to toxic metals is known to negatively affect plant traits that are important for plant-water relationships, and observed changes in the case of drought and heavy metal stress are similar (de Silva et al., 2012; Rucińska-Sobkowiak 2016). Due to the effect of Cd, tissue dehydration also occurred in our study (Figure 2), therefore we assume that the observed changes may be an indirect consequence of the effect of Cd. A new citation has been added to the manuscript. Line
de Silva ND, Cholewa E, Ryser P.: Effects of combined drought and heavy metal stresses on xylem structure and hydraulic conductivity in red maple (Acer rubrum L.). J Exp Bot. 2012 63 (16), 5957-66. doi: 10.1093/jxb/ers241.
Comment:
There is a lack about the investigation of transporters of cadmium at the level of gene expression. It would be nice to see how these genes are involved in these responses.
Response 4: Thank you for your comment. The importance of some transporters in the process of Cd uptake and translocation is mentioned in the manuscript. Some aspects of Cd transport at the level of expression were detailed reported (Thomine et al., 2000; Yu et al., 2017, Zhang et al., 2018a; Zhang et al., 2018b; Jian et al., 2019).
Thomine S, Wang R, Ward JM, Crawfordb NM, Schroeder JI: Cadmium and iron transport by members of a plant metal transporter family in Arabidopsis with homology to Nramp genes, Proc Natl Acad Sci USA, 2000; 97(9):4991-6.
Yu R, Li D, Du X, Xia S, Liu C, Shi G.: Comparative transcriptome analysis reveals key cadmium transport-related genes in roots of two pak choi (Brassica rapa L. ssp. chinensis) cultivars, BMC Genomics 2017, 18, Article no. 587.
Zhang J, Martinoia E, Lee, Y.: Vacuolar transporters for cadmium and arsenic in plants and their applications in phytoremediation and crop development. Plant and Cell Physiology 2018a, 59 (7), 1317–1325.
Zhang XD, Meng JG, Zhao KX, Chen X, Yang ZM: Annotation and characterization of Cd-responsive metal transporter genes in rapeseed Brassica napus. Biometals 2018b, 31, 107–21.
Jian S, Luo J, Liao Q, Liu Q, Guan C, Zhang Z.: NRT1.1 Regulates nitrate allocation and cadmium tolerance in Arabidopsis. Frontiers in plant science 2019, 10, 384.
Further analyzes within this study were not possible due to lack of material (seeds- Vicia faba, Aštar variety). Seeds of the variety became unavailable with the interval of experiments. Due to the variability in the reaction of different genotypes to cadmium ions, the use of a different variety could be confusing. Nevertheless, we are convinced that the presented results summarized in the submitted manuscript could be beneficial for the scientific community, and we therefore decided to publish them.
Reviewer 3 Report
A manuscript entitled “Effect of cadmium chloride and cadmium nitrate on growth and mineral nutrient content in the root of fava bean (Vicia faba L.)” is well written. While it is informative and of value to researchers. However, the manuscript needs substantial improvements. The novelty of work should be clearly mentioned. Suggestions as follows!
Abstract: A concluding point regarding the importance of investigation should be presented in the last.
Introduction: A line on heavy metal contamination with respect to plant growth and development is needed
Why faba been was selected for this investigation? Some economic importance of this plant needs inclusion. Why Cadmium was selected for the investigation should be explained for the same reason?
Line 64: “up-regulate” to “upregulate”
The major objective needs some more explanation to justify the conducted work and must point out the significance of work done in the perspectives of agricultural productivity/crop yield.
Results:
Abbreviations like FW, DW, and TI should be explained at the first mentioned places.
Figure 2 caption: Why comma “,” instead of full stop“.” For indicating p<0.05. Please check throughout the manuscript.
Line 158: To which numerical value of correlation co-efficient authors considered high positive correlation? Please explain.
Discussion:
Line 186: Please include the scientific name
Line 194: If seeds are impermeable to heavy metals, how did metal solution affect the growth? In my opinion, this should be mentioned as less permeable rather than impermeable. Please consider and amend.
Line 206: “Elongation” only instead of “elongation growth”
Line 253: What is the effect of Cd(NO3)2 treatment?
Line 256: Please include the chemical similarity between Ca and Cd.
Please include a concluding point of all the discussion in the last.
Line 310: Were all the taken roots had constant weights? There may be differences and may influence the sample absorbance.
Section 4.5: Please mention the limit of detection (LOD) and limit of quantification (LOQ) for each of the elements analyzed. The name of the instrument should be indicated with the country of origin throughout the manuscript. The name of standard reference material also needs representation at the specified place.
Authors should not add results and discussion in conclussion section. It should be concise and informative.
Authors should avoid to use words: our experiments, we did, etc. throughout the manuscript.
Author Response
Dear Reviewer 3.
Thank you for the comments. I am sending you the revised version of the manuscript entitled “Effect of cadmium chloride and cadmium nitrate on growth and mineral nutrient content in the root of fava bean (Vicia faba L.).” We have considered all comments of the reviewers in the manuscript. All of modifications were marked by red color in revised manuscript. However, if there is any other comment for improving the MS I would be much appreciated to know.
Best Regards,
Emília Ondrušková
Response to reviewer 3:
Comment: Abstract: A concluding point regarding the importance of investigation should be presented in the last.
Response 1: The Abstract was corrected according to the reviewers´ comments; concluding point was added.
Comment: Introduction: A line on heavy metal contamination with respect to plant growth and development is needed
Why faba been was selected for this investigation? Some economic importance of this plant needs inclusion. Why Cadmium was selected for the investigation should be explained for the same reason?
Response 2: The required information and literature has been added to the Introduction. A new citation has been added to the manuscript. Lines 49-56.
Tang L, Hamid Y, Zehrab A, Sahito ZA, He Z, Hussaina B, Gurajala HK, Yang X.: Characterization of fava bean (Vicia faba L.) genotypes for phytoremediation of cadmium and lead co-contaminated soils coupled with agro-production. Exotoxicol. Environ. Saf. 2019, 171, 190-198. doi: 10.1016/j.ecoenv.2018.12.083
Comment: Line 64: “up-regulate” to “upregulate”
Response 3: Corrected: "upregulate" instead of "up-regulate"
Comment: The major objective needs some more explanation to justify the conducted work and must point out the significance of work done in the perspectives of agricultural productivity/crop yield.
Response 4:The required has been added in the manuscript. Lines 49-56, 82-86.
Comment: Abbreviations like FW, DW, and TI should be explained at the first mentioned places.
Response 5: Abbreviations FW, DW, and TI were explained at the first mentioned places in the manuscript. Lines 115 , 118
Comment: Figure 2 caption: Why comma “,” instead of full stop“.” For indicating p<0.05. Please check throughout the manuscript.
Response 6: The mistake (0.05 instead of 0,05) was corrected throughout the manuscript.
Comment: Line 158: To which numerical value of correlation co-efficient authors considered high positive correlation? Please explain.
Response 7: Numerical values indicating high positive correlation between parameters were added: „Increasing the doses of cadmium chloride and cadmium nitrate caused the gradual increase in the content of Cd (r = 0.97, r = 0.99, p <0.5), Ca (r = 0.95, r = 0.98, p <0.5) and Mg in the roots (r = 0.99, r = 0.98, p <0.5).“ Lines 164 -170.
Comment: Line 186: Please include the scientific name
Response 8: The scientific name of turnip rape (Brassica campestris) was added. Line 194.
Comment: Line 194: If seeds are impermeable to heavy metals, how did metal solution affect the growth? In my opinion, this should be mentioned as less permeable rather than impermeable. Please consider and amend.
Response 9: The sentence was corrected as recommended by reviewers. Line 202.
Comment: Line 206: “Elongation” only instead of “elongation growth”
Response 10: The words were rewritten according to the instructions of the reviewer. Line 213.
Comment: Line 253: What is the effect of Cd(NO3)2 treatment?
Response 11: I'm not sure I understand the question well. Effect of Cd(NO3)2 was disscused in Lines192-216.
Comment: Line 256: Please include the chemical similarity between Ca and Cd.
Response 12: The chemical similarity between Ca and Cd was included in the text: „Due to the chemical similarity between Ca and Cd (similar charge and ionic radius),.......“ Line 264.
Comment: Please include a concluding point of all the discussion in the last.
Response 13: The concluding point of all the discussion was aded in the last.
Comment:
Line 310: Were all the taken roots had constant weights? There may be differences and may influence the sample absorbance.
Response 14: I agree, explanation added in section 4.4., the roots had constant weight 300 mg. Line 328.
Comment:
Section 4.5: Please mention the limit of detection (LOD) and limit of quantification (LOQ) for each of the elements analyzed. The name of the instrument should be indicated with the country of origin throughout the manuscript. The name of standard reference material also needs representation at the specified place.
Response 15: The limits of detection (LOD) and limits of quantification (LOQ) for each analysed minerals were added. Data about used instruments and standard reference materials were added. Lines 349 – 356.
Comment:
Authors should not add results and discussion in conclussion section. It should be concise and informative.
Response 16: The Conclusions chapter was rewritten according to the reviewers' comments.
Comment:
Authors should avoid to use words: our experiments, we did, etc. throughout the manuscript.
Response 17: The manuscript was revised and the passive voice was used.
Round 2
Reviewer 2 Report
Authors almost accepted and made all the revisions requested, however in the future I recommend more robust approach to study the stress responses. Minor comment: scale bar is missing from the Fig.1.
Author Response
Once again, we would like to thank all the reviewers for their valuable comments and advice that have helped to improve the manuscript. The recommendations will help us in further planning and observations.
Authors almost accepted and made all the revisions requested, however in the future I recommend more robust approach to study the stress responses. Minor comment: scale bar is missing from the Fig.1.
Response 1: The Scale bare was added to Fig.1.
Reviewer 3 Report
The authors responded to all queries.
Author Response
Once again, we would like to thank all the reviewers for their valuable comments and advice that have helped to improve the manuscript. The recommendations will help us in further planning and observations.